# Knowledge, Practice, Compliance, and Barriers toward Ventilator-Associated Pneumonia among Critical Care Nurses in Eastern Mediterranean Region: A Systematic Review

**DOI:** 10.3390/healthcare10101852

**Published:** 2022-09-23

**Authors:** Khaild AL-Mugheed, Wegdan Bani-Issa, Mohammad Rababa, Audai A. Hayajneh, Adi Al Syouf, Mohammad Al-Bsheish, Mu’taman Jarrar

**Affiliations:** 1Faculty of Nursing, Surgical Nursing Department, Near East University, Nicosia 99138, Cyprus; 2College of Health Science\Nursing Department, University of Sharjah, Sharjah 26666, United Arab Emirates; 3Department of Adult Health-Nursing, Faculty of Nursing, Jordan University of Science and Technology, Irbid 22110, Jordan; 4Department of Managing Health Services and Hospitals, Faculty of Business Rabigh, College of Business (COB), King Abdulaziz University, Jeddah 21991, Saudi Arabia; 5Health Management Department, Batterjee Medical College, Jeddah 21442, Saudi Arabia; 6Al-Nadeem Governmental Hospital, Ministry of Health, Amman 11118, Jordan; 7Medical Education Department, King Fahd Hospital of the University, Al-Khobar 34445, Saudi Arabia; 8Vice Deanship for Quality and Development, College of Medicine, Imam Abdulrahman Bin Faisal University, Dammam 34212, Saudi Arabia

**Keywords:** ventilator-associated pneumonia, critical care nurses, eastern Mediterranean region, knowledge, practice, compliance

## Abstract

Background: Ventilator-associated pneumonia (VAP) has been identified as a serious complication among hospitalized patients and is associated with prolonged hospitalizations and increased costs. The purpose of this study was to examine the knowledge, practices, compliance, and barriers related to ventilator-associated pneumonia among critical care nurses in the eastern Mediterranean region. Methods: The PRISMA guidelines guided this systematic review. Four electronic databases (EMBASE, MEDLINE (via PubMed), SCOPUS, and Web of Science) were used to find studies that were published from 2000 to October 2021. Results: Knowledge of ventilator-associated pneumonia was the highest outcome measure used in 14 of the 23 studies. The review results confirmed that nurses demonstrated low levels of knowledge of ventilator-associated pneumonia, with 11 studies assessing critical care nurses’ compliance with and practice with respect to ventilator-associated pneumonia. Overall, the results showed that most sampled nurses had insufficient levels of compliance with and practices related to ventilator-associated pneumonia. The main barriers reported across the reviewed studies were a lack of education (N = 6), shortage of nursing staff (N = 5), lack of policies and protocols (N = 4), and lack of time (N = 4). Conclusions: The review confirmed the need for comprehensive interventions to improve critical care nurses’ knowledge, compliance, and practice toward ventilator-associated pneumonia. Nurse managers must address barriers that impact nurses’ levels of knowledge, compliance with, and practices related to ventilator-associated pneumonia.

## 1. Introduction

Ventilator-associated pneumonia (VAP) is the most common nosocomial pneumonia, occurring two and three days following endotracheal intubation [1]. VAP is distinguished by having a new progressive infiltrate, alteration in sputum characteristics, high white blood cell count, and high body temperate [2]. The first five days of intubation are considered the highest risk incidence of VAP, and the mean period from the time of intubation to the development of VAP is four days [3].

The literature has demonstrated that VAP is associated with increased mortality and morbidity, prolonged hospitalization, and increased care costs [4]. In a one-year period, the median total cost of VAP patients in developing countries was 6308 Euros compared to 2315 Euros in non-VAP patients [5]. In developing countries with limited healthcare resources and supplies, the incidence rate of VAP is higher than in developed countries [6]. A recent study in Egypt reported that early-onset VAP was 44%, and late-onset was 56% [7]. In Iran in 2016, the incidence of VAP in the control group was 55.3% which was higher than in the intervention group at 27.5% [8]. In Jordan, the mortality rate was 46.4% among VAP patients [9].

Device-related, personnel-related, and host-related factors can increase the chance of a patient developing VAP [10,11]. Device-related factors include reintubation after extubation, ventilation circuit, endotracheal tube, and an orogastric tube. Personnel-related factors include a lack of use of personal protective equipment and incorrect hand hygiene. Host-related factors include underlying diseases, aging, a low level of consciousness, cardiovascular system diseases, and antibiotic medications [12,13]. 

Evidence shows that implementing a VAP prevention bundle provides trustworthy directives and an effective decrease in VAP rates, improving patient safety, and quality of care [14,15,16]. Many organizations, including the European Respiratory Society [10], The Society for Healthcare Epidemiology of America [11], the Intensive Care Society [17], the American Thoracic Society [18], the Center for Disease Control and Prevention [19], and the Institute for Health Care Improvement [20] have launched a “ventilator information bundle” to decrease VAP mortality and increase its prevention. These informational bundles include clinical practice guidelines such as oropharyngeal hygiene, suction endotracheal secretions, elevation of the head at 30–45 degrees, oral care with chlorhexidine, daily sedation interruption, subglottic secretion drainage, hand hygiene, assessing the cuff pressure of the endotracheal tube, and facilitating early mobilization [10,11,17,18,19,20]. 

Several studies have demonstrated that nurses have insufficient knowledge and often do not engage in practices which can prevent VAP among intensive care unit (ICU) patients. In a recent study conducted in Iran examining emergency nurses’ knowledge regarding VAP prevention, participants showed inadequate knowledge of how to prevent VAP [14]. In a study conducted in Turkey, nurses working in ICUs revealed poor VAP knowledge [21], and a study from Yemen showed that nurses working in ICUs had poor knowledge regarding VAP prevention [22]. In an observational study, Jordanian nurses showed insufficient compliance” regarding VAP prevention guidelines [23].

Although the barriers that impede nurses from implementing VAP prevention practices have not been fully explored yet, a lack of time and workload, lack of education, shortage of staff, and lack of policies and protocols are most frequently reported and may contribute to high prevalence of VAP [24,25]. In a study conducted in Jordan to examine mechanical ventilator and oral care practice among Jordanian nurses, the nurses reported low quality of oral care and lack of compliance [26]. In another study conducted in (N = 4) teaching hospitals in Syria to examine the effectiveness of a VAP prevention bundle on compliance and the rate of VAP, failure of applied bundle due to inconsistent implementation of the bundle between staff was reported [27]. A self-reported questionnaire was completed by health-care providers. The study noted significant difficulties related to the lack of policies such as the presence of many doctors responsible for patient care and the difference in VAP implementation between different units within participating hospitals [28]. This systematic review examines knowledge, practices, compliance levels, and investigates barriers to VAP prevention among critical care nurses in the eastern Mediterranean region. 

## 2. Materials and Methods

### 2.1. Information Sources

The literature search for this systematic review was adopted based on the Preferred Reporting Items for Systematic Reviews and Meta-Analyses checklist (PRISMA) guidelines [29]. Four online electronic databases, EMBASE, MEDLINE (via PubMed), SCOPUS, and Web of Science, were searched to identify relevant full-text studies in humans published between 2000 and October 2021. The first step was to search via online electronic databases and then analyze the article’s title and abstract in-text words. Then, a second search was performed using all selected keywords and index terms across all of the listed databases. Finally, the reference lists of previous studies were examined for further relevant articles, and keywords were combined with Boolean operators, including AND and OR.

### 2.2. Inclusion and Exclusion Criteria

All published studies that examined knowledge, practice, compliance, and barriers toward VAP prevention guidelines among critical care nurses in the eastern Mediterranean region were included in this review. The eastern Mediterranean region is defined based on the WHO category [30]. (See http://www.emro.who.int/countries.html (accessed on 10 march 2020).Also, studies that used the cross-sectional design, randomized controlled trial or quasi-experimental study design, pre-post-test design, and self-report or observation method for data collection were included. All published studies that used several types of participant groups such as healthcare providers or students, unpublished graduate theses, review articles, case studies, conference abstracts, studies with low quality, and studies not in English were excluded.

### 2.3. Search Strategy

Before the process, a health sciences librarian was consulted regarding the search methodology. Indexing terms included ((“Intensive Care Units” [Mesh]) OR “Critical Care Nursing” [Mesh])) AND “VAP, Knowledge” [Mesh] in MEDLINE, ((“Critical Care Nursing” Mesh]) OR “Intensive Care Nursing” [Mesh])) AND “Ventilator-Associated Pneumonia, Practice,” in Web of Science, while ‘Intensive Care’ /exp AND ‘Ventilator-Associated Pneumonia’ AND ‘Barrier’ in Scopus, ‘Critical Care Nursing’/de [tiab] OR ‘Intensive Care Nursing’ [tiab] AND ‘Ventilator-Associated Pneumonia’ [tiab] AND ‘Compliance’ [tiab] in EMBASE. The search strategy was used in three-step to identify primary studies on ICU nurses and VAP Table 1. 

### 2.4. Study Identification

In the filtration and screening phase, two researchers independently filtered for duplicates of the titles and abstracts of all studies. After removing duplicates, the independent authors performed a screen for potential relevance to the eligibility criteria and coded it as “cover”. After independent screening, disagreements were resolved by reaching a consensus. A third independent reviewer was consulted to resolve the disagreement if a consensus could not be reached. Studies with appropriate data were included in the systematic review. The required data extracted included study characteristics (year of publication, data collection method, participants, sampling method) and levels of nurses’ knowledge, practice, and compliance with VAP. 

### 2.5. Risk of Bias

The Office of Health Assessment and Translation (OHAT) tool was used to assess the risk of bias [31]. The OHAT tool was adopted according to CLARITY Group guidance at McMaster University. The OHAT tool includes six domains (selection, confounding, performance, attrition/exclusion, detection, and selective reporting). The risk-of-bias ratings for each domain four answers “definitely low,” “probably low,” “probably high,” and “definitely high.” Two authors assessed each paper for risk of bias, and discussions resolved any discrepancies.

## 3. Results

### 3.1. Study Selection

A total of 2520 articles were identified from the initial search in 4 online electronic databases. After the deletion of duplicates, 1719 studies were addressed for further screening. After reading the abstract and full texts, 1625 articles were excluded. Full-text articles were assessed for eligibility, and 94 articles were included. A total of 71 articles were excluded for several reasons, including studies not conducted in the Eastern Mediterranean region (49 articles), mixed populations (15 articles), written in languages rather than English (6 articles), and conducted on nursing students (1 article). Finally, according to the independent researcher’s agreement, 23 studies were included in the final review (Figure 1).

### 3.2. Study Characteristics

Twenty-three studies conducted on 3841 critical care nurses were included in this study. Most studies were conducted in Jordan (N = 7). [23,24,25,32,33,34,35], Iran (N = 6) [36,37,38,39,40,41], Saudi Arabia (N = 4) [34,42,43,44], and Egypt (N = 3) [43,45,46]. Of the 23 studies, 11 did not use a sampling method, and six studies used a convenience sampling method [25,43,46,47,48,49]. Self-reporting and self-administrated methods were used to complete the questionnaires in most studies. More than half of the studies used a cross-sectional, descriptive study design (N = 13). See Table 2 for further details.

### 3.3. Used Instruments

Of the 23 studies, 10 studies used standardized instruments and mentioned the number of instrument items [22,23,24,25,32,33,34,40,44]. Two studies used probing questions and open-ended interview questions [36,38]. Two studies had the lowest number of instrument items (N = 9) in terms of VAP knowledge [39,48], while the study of Al Shameri (2017) had the highest number of instrument items (N = 40) [50]. In terms of compliance and practice, Aloush’s (2017) study included the lowest number of instrument items (N = 8) [23], while two studies included the highest number of instrument items (N = 17) [40,44]. In terms of instruments’ psychometric properties, 11 studies reported the validity score of the instrument. The reported Cronbach’s alphas in the reviewed studies ranged from 0.69 to 0.92. See Table 2 for further details.

### 3.4. Nurses’ General Knowledge of VAP

The knowledge of VAP was the highest measure used in 14 studies of the 23 studies. In most studies, the knowledge level was classified as low, inadequate, significant improvement after education, or poor. Only one study reported that nurses had “adequate knowledge” [24]. Of the 14 studies, 4 studies reported nurses’ level of knowledge as “low”, 3 studies characterized the level of knowledge as “inadequate,” and another 3 studies mentioned that nurses had “significant improvement after education”. The remaining two studies reported that nurses had “poor knowledge”. In the last remaining study, nurses had “unsatisfactory knowledge”. The overall results showed that nurses had a low VAP knowledge level. See Table 2 for further details.

### 3.5. Nurses’ Compliance and Practice of VAP

Of the 23 studies, 11 assessed critical care nurses’ compliance with VAP practices. Critical care nurses’ practice and compliance levels were classified as high, insufficient, and acceptable. Only one study mentioned that nurses had a “high level” of compliance and practice of VAP [32]. Of the 11 studies, 4 reported participants’ insufficient compliance and practice of VAP, and in 2 studies, the participants had an “acceptable level.” The overall results showed that most nurses had insufficient compliance with and knowledge of VAP practices. See Table 2 for further details.

### 3.6. Quality of Studies

In general, the majority of studies reported as having a high quality level with low bias. The most commonly found terms “Definitely High” and “Probably High” were related to the confounding, performance and attrition/exclusion. Of the 23 studies, 4 studies reported “Definitely High” in terms of confounding [26,39,42,43]. Two studies reported “Probably High” related to performance [28,43]. See Table 3 for further details. 

### 3.7. Barriers to Adherence to VAP Guidelines

Nine studies assessed barriers to adherence to VAP guidelines of the overall included studies. The most-reported barriers in several studies were (1) lack of education (N = 6), (2) shortage of nursing staff (N = 5), (3) lack of policies and protocols, and lack of time (N = 4). Table 4.

## 4. Discussion

This systematic review examines knowledge, practices, compliance levels, and investigates barriers to VAP prevention among critical care nurses in the eastern Mediterranean region. The overall results showed that nurses in the eastern Mediterranean region had a low level of knowledge of VAP prevention than critical care nurses in Europe, in which studies reported that the nurses had adequate knowledge of VAP prevention [51,52]. The possible reason for this variance may be attributed to differences between education systems and institutional policies. Intervention studies are efficient for improving nurses’ VAP knowledge and considering vital for a significant decrease in VAP incidence and the cost of treatment [33,35]. For example, in a prospective cohort study conducted in 5 different countries among 44 ICUs to examine the effect of multidimensional sessions, the study showed that VAP rates were significantly reduced after implementing the training and educational sessions [53]. The level of nurses’ knowledge was raised after the implementation of the training, as they were enabled to identify the correct evidence-based VAP preventive measures [52]. Additionally, after the training, nurses were able to identify which preventive measure priorities are classified as highly, moderately, and less recommended based on VAP guidelines [21].

Of the 11 studies, only one mentioned that nurses had high compliance and practice levels related to VAP [32]. Higher adherence to VAP bundles may present challenges outside a nurse’s control, such as gaining resources, continuing education, and observation schedules [25]. In addition, applying active strategies, such as incentives, support for the decision, regular observation, and assessing bundle issues might be more cost-effective than appropriate to encourage nurses to perform and adhere to the bundle [52]. Reinforcement of compliance and practice of VAP should be conducted regularly and evaluated for proper performance during their clinical work.

The current study observed variability in the study design, instrument standards, validity, and data collection methods. The cross-sectional, descriptive design was utilized in more than half of the reviewed studies. Although these designs are quick and inexpensive to conduct, they may demonstrate several challenges, such as difficulty interpreting relationships identified and making a causal inference [54,55,56,57]. Using a qualitative or mixed-method design would have given in-depth and more specific responses from the participants than the self-reported instruments used [58,59,60]. Furthermore, qualitative design may be more convenient for focusing on barriers faced by nurses and how nurses’ knowledge influences their compliance toward using VAP bundles in clinical practice [36]. Most instruments were adopted for measuring variables knowledge, compliance and practice of VAP, but there have often been many adaptations and replications of the same instrument. These frequent adaptations and replications may limit the clarity of the instrument construct by retaining the original name and using the instrument without re-validation. Eleven studies did not examine the instrument’s validity. These findings were similar to international studies that do not report the reliability and validity of VAP instruments [61,62,63]. This indicates that survey results may not be consistent as various instruments were utilized to examine the same variables. The strength of instruments depends on the magnitude and their psychometric test scores [64].

The majority of studies used self-report and self-administrated methods for measuring their variables. Although self-report and self-administrated questionnaires are useful with a large sample and are inexpensive, their response rates can be low [54]. Previously published studies have demonstrated that the self-report method has several limitations, such as exaggeration of compliance rate, recall biases, high floater answers, and acquiescence or agreement bias [25,35]. Using direct observation as a collecting data method with validated instruments may lead to a more precise evaluation of VAP practices [22]. A convenience sampling method was used in the majority of the selected studies. Convenience sampling may yield difficulty generalizing the findings and have limited external validity [65,66,67]. Future studies should use rigorous sampling from a large population.

Several recommendations are mentioned for the reviewed studies. The lack of education was the most reported barrier to optimal management of VAP, highlighting a significant weakness in nursing curricula (as reported in prior studies) [35,68]. These studies indicated that the lack of in-service education could have increased the risk of VAP complications. Continuous education has been strongly recommended as the cornerstone of nurses’ knowledge and compliance improvement for VAP management [23]. **In this study, nurses also reported that shortage of nursing staff and lack of policies and protocols as the most common barriers encountered in clinical practice settings. Having these kinds of barriers might cause a lack of compliance. This finding was consistent with studies noted that developing policies and protocols to standardize the implementation of a bundle was a substantial factor in implementing a bundle [23,68].**

### Limitations

The main methodological shortcoming in this study is the inadequate number of covered studies. Another limitation is excluding published studies in languages other than English, which may introduce a selection bias. The studies have not discussed how to address common method bias. Common method bias can be reduced by the utilization of several data collection methods, number of items, validity of contents, and times and locations required to obtain accurate results. However, this exclusion was applied to ensure that identified studies had high quality and integrity. The selected studies were conducted only in the Eastern Mediterranean region, which yielded only seven countries, and this may limit the generalizability of the results.

## 5. Conclusions

This is the first systematic review conducted to examine knowledge, practices, compliance levels and investigate barriers to VAP prevention among critical care nurses in the eastern Mediterranean region. The review results confirmed that nurses in the eastern Mediterranean region showed low levels of knowledge and insufficient levels of compliance of ventilator-associated pneumonia. This indicated that nurse managers and policymakers take on their considerations enhancing nurses’ preparation for working in critical care units. Although several barriers might interfere with nurses’ best practice such as a lack of education, shortage of nursing staff, lack of policies and protocols, and lack of time, applying tailored educational programs may help improve nurses’ knowledge and compliance and help eliminate these barriers.

## Figures and Tables

**Figure 1 healthcare-10-01852-f001:**
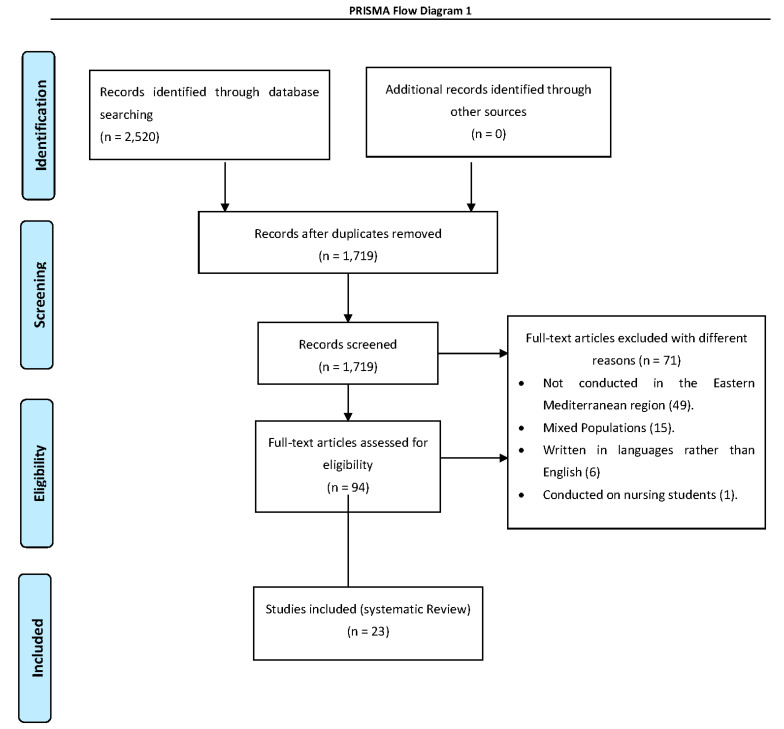
PRISMA Flow Diagram.

**Table 1 healthcare-10-01852-t001:** List of terms used and search results.

No.	Database	Terms	Search Result
1	Pubmed (MeSH)	(“Intensive Care Units” [Mesh] OR “Critical Care Nursing” [tiab]) AND (“knowledge” [Mesh] [tiab] OR “Practice” [tiab]) AND (“Ventilator-Associated Pneumonia” [tiab] OR “VAP” [tiab]).	1135
2	EMBASE (emtree)	(“Nursing” [Mesh] OR “Critical Care”[tiab] OR “Intensive Care” [tiab])AND (“adherence” [Mesh] “[tiab] OR” Compliance [tiab]) AND (“Ventilator-Associated Pneumonia” [tiab] OR “VAP” [tiab]).	299
3	Web of Science	“Critical Care Nursing” [Mesh] OR “Critical Care Nurses” [tiab] OR “intensive care” [tiab]) AND (“practice” [tiab] OR “skills” [tiab] AND (“Ventilator-Associated Pneumonia” [tiab] OR “Healthcare-Associated Pneumonia” [tiab]) OR “Ventilator-Associated”[tiab]).	338
4	SCOPUS	“Critical Care Nursing” “[Mesh] OR “Critical Care Nurses” [tiab] OR “intensive care” [tiab]) AND (“barriers” [tiab] OR “obstacles” [tiab] OR “challenges” [tiab] OR “difficulties” [tiab] “issues” [tiab]) AND (“Ventilator-Associated Pneumonia” [tiab] OR “Healthcare-Associated Pneumonia” [tiab]) OR “Ventilator-Associated” [tiab]).	748

**Table 2 healthcare-10-01852-t002:** Study characteristics.

Study Authors (Year)	Outcome Measures	Country	Study Characteristics	Study Design	Instrumentation	Main Outcomes
Darawad et al. [32], 2018	Knowledge and Practices	Jordan	Participants: 208 Target Population: Intensive Care Unit Nurses. Sampling Method: A stratified random sampling. Method of Data Collection: Self-Reported	Experimental	Type of tool: Standardized. Number of items: 10 items for knowlgde,15 items for practice Reliability: NA	Poor knowledge and high practices
Hamishehkar, et al. [37], 2014.	Compliance	Iran	Participants: 143. Target Population: Intensive Care Units Nurses Sampling Method: NA Method of Data Collection: Three steps; first step; VAP care bundle compliance, second and third steps pre and post education.	Observational	Type of Tool: Non-Standardized. Number of items: NA. Reliability: NA.	Insufficient Compliance
Aloush, et al. [34], 2017.	Compliance and Barriers	Jordan, Egypt, and Saudi Arabia	Participants: 471. Target Population: Intensive Care Unit Nurses. Sampling Method: convenience sample. Method of Data Collection: Self-reported.	Observational	Type of Tool:Standardized. Number of items: 9 items compliance, 15 items related to barriers. Reliability: Cronbach α, 0.8.	Insufficient Compliance. Lack of education. Lack of a professional model. Poor integration of research findings in practice
Al-Sayaghi et al. [44], 2014	Knowledge	Yemen	Participants: 387. Target Population: Intensive Care Unit Nurses. Sampling Method: NA Method of Data Collection: self administered	Observational	Type of Tool: Standardized. Number of items: 15 items Reliability: NA	Low Knowledge
Atashi et al. [26], 2018	Barriers	Iran	Participants: 23 Target Population: Critical care nurses Sampling Method: A purposive sample Method of Data Collection: Semi-structured interviews.	Qualitative	Type of Tool: NA Number of items: 3 probing questions Reliability: NA	Lack of education. Lack of a professional model. Unfavourable environmental conditions. Shortage of nursing staff. Lack of time and resources Passive human resource and organizational management
Rashnou et al. [38], 2017	Barriers	Iran	Participants: 12. Target Population: Critical care nurses Sampling Method: A purposive sample Method of Data Collection: Semi-structured interviews	Qualitative	Type of Tool: NA Number of items: 5 Broad open-ended interview questions. Reliability: NA	Unfavourable environmental conditions. Passive human resource and organizational management
Yaseen and Salameh, [43], 2015	Knowledge, Barriers	Saudi Arabia	Participants: 93 Target Population: Critical care nurses Sampling Method: NA Method of Data Collection: Self-Reported	Observational	Type of Tool: Non-Standardized. Number of items: 11 items, nine questions in terms of barriers Reliability: NA.	Low Knowledge. Lack of education. Shortage of nursing staff. Lack of policies and protocols.
Aloush, [23], 2017	Compliance.	Jordan	Participants: 100 Target Population: Critical care nurses Sampling Method: NA Method of Data Collection: non-participant observers	Observational	Type of Tool: Standardized. Number of items: 8 items Reliability: Cronbach’s alpha of 0.8.	Insufficient Compliance
Tabaeian et al. [40], 2017	Compliance	Iran	Participants: 120 Target Population: Critical care nurses Sampling Method: NA Method of Data Collection: observation using a checklist	Observational	Type of Tool: Standardized. Number of items: 17 items Reliability: Cronbach α, 0.698	Acceptable Compliance
Aloush SM, [33], 2017	Compliance	Jordan	Participants: 102 Target Population: Critical care nurses Sampling Method: random sample (experimental group and the control group). Method of Data Collection: education and observation	Quasi-experimental	Type of Tool: Standardized. Number of items: 9-items Reliability: NA	Moderate Compliance. There was no statistically significant difference between experimental group and the control group
Al-Shameri FA, [50], 2017	Knowledge	Sudan	Participants: 120 Target Population: Critical care nurses Sampling Method: non-probability, purposive sample Method of Data Collection: Self-administrated	Observational	Type of Tool: Standardized. Number of items: 40-items Reliability: Cronbach α, 0.87	Inadequate Knowledge.
Al-Sayaghi KM, [44], 2020	Compliance and Barriers	Saudi Arabia	Participants: 229 Target Population: Critical care nurses Sampling Method: NA Method of Data Collection: Self-administrated	Observational	Type of Tool: Standardized. Number of items: 17 items for compliance, and 15 items for barriers Reliability: Cronbach α, 0.79	Acceptable Complianc. Shortage of nursing staff. Forgetfulness Lack of policies and protocols.
Hussein et al. [47], 2020	Knowledge	Iraq	Participants: 126 Target Population: Critical care nurses Sampling Method: convenience sample Method of Data Collection: Self-Reported	Observational	Type of Tool: None Standardized. Number of items: 20 items Reliability: NA	Poor knowledge
Al-khazali et al. [24], 2021	Knowledge and Barriers	Jordan	Participants: 185 Target Population: Critical care nurses Sampling Method: NA Method of Data Collection: Self-Reported	Observational	Type of Tool: Standardized. Number of items: 30 questions items for knowledge, and 8 items for barriers Reliability: Cronbach α, 0.77.	Adequate Knowledge. Lack of education. Shortage of nursing staff. Forgetfulness
Khalifa et al. [45], 2020	Knowledge and Practices	Egypt	Participants: 70 Target Population: Critical care nurses Sampling Method: NA Method of Data Collection: Education (Pre- posttest).	A quasi- experimental	Type of Tool: None Standardized. Number of items: 15 questions items for knowledge, and 10 items for practices. Reliability: NA	Significant improvement after education in terms of knowledge and practices
Al-jaradi et al. [49], 2020	Knowledge	Yemen	Participants: 205 Target Population: Critical care nurses Sampling Method: A convenience sampling Method of Data Collection: Self-Reported	Observational	Type of Tool: None Standardized. Number of items: 28 questions items. Reliability: 0.73	Inadequate knowledge
Yeganeh et al. [41], 2016	Knowledge and Barriers	Iran	Participants: 219 Target Population: Critical care nurses Sampling Method: NA Method of Data Collection: Self-Reported	Observational	Type of Tool: None Standardized. Number of items: 9 items for knowledge, and 11 items for barriers Reliability: NA.	Inadequate knowledge. Lack of education.
Nahla SA., [46], 2013	Knowledge and Compliance	Egypt	Participants: 45 Target Population: Critical care nurses Sampling Method: sample of convenience Method of Data Collection: Self-administered and observational checklist	Observational	Type of Tool: None Standardized. Number of items: 20 items for knowledge, and 12 items for Compliance Reliability: 0.87	Unsatisfactory knowledge and insufficient compliance
Hawsawi et al. [42], 2018	knowledge and practices	Saudi Arabia	Participants: 109 Target Population: Critical care nurses Sampling Method: Convenience sample Method of Data Collection: Education (Pre- posttest), Observation	Quasi experimental	Type of Tool: None Standardized. Number of items: 32 items for knowledge, and 10 items for practices Reliability: Cronbach α, 0.82	Significant improvement after education interms of knowledge and practices
Hassan and Wahsheh, [26], 2017	Knowledge and Barriers	Jordan	Participants: 428 Target Population: Critical care nurses Sampling Method: stratified sample Method of Data Collection: self-administered and pre-intervention, (b) educational and (c) post-intervention.	Quasi experimental	Type of Tool: Standardized. Number of items: 20 items for knowledge, and one open end question for barriers Reliability: Cronbach α, 0.78	Significant improvement after education interms of knowledge. Lack of policies and protocols. Lack of time
Aloush and Al-Rawajfa, [19], 2020	Compliance and Barriers	Jordan	Participants: 294 Target Population: Critical care nurses Sampling Method: convenience sample Method of Data Collection: Self-reported	Observational	Type of Tool: Standardized. Number of items: 10 items for knowledge, and 15 items for barriers Reliability: Cronbach α, 0.82	Poor Compliance. Lack of education. Shortage of nursing staff. Lack of policies and protocols.
Bagheri et al. [30], 2013	Knowledge	Iran	Participants: 52 Target Population: Critical care nurses Sampling Method: NA Method of Data Collection: Self-Reported	Observational	Type of Tool: None Standardized. Number of items: 9 items Reliability: Cronbach α, 0.92	Low Knowledge
Zeb et al. [39], 2018	Knowledge	Pakistan	Participants: 100. Target Population: Critical care nurses Sampling Method: Convenience Method of Data Collection: Self-Reported	Observational	Type of Tool: None Standardized. Number of items: 15 items Reliability: NA	Low Knowledge

**Table 3 healthcare-10-01852-t003:** Quality of studies.

	Selection	Confounding	Performance	Attrition/Exclusion	Detection	Selective Reporting
Darawad et al., 2018	DL	DL	DL	DL	PL	PL
Hamishehkar, et al., 2014	PL	DL	DL	DL	DL	PL
Aloush, et al., 2017.	PL	PL	DL	DL	DL	DL
Atashi et al., 2018	DL	DH	PL	PL	PL	PL
Rashnou et al., 2017	DL	DL	PH	DL	DL	PL
Al-Sayaghi et al., 2014	PL	DL	DL	PL	PL	DL
Aloush, 2017	DL	DL	DL	DL	DL	DL
Yaseen and Salameh, 2015	DL	DH	PH	PH	PL	PL
Al-Oush SM, 2017	DL	DL	DL	DL	DL	DL
Al-Shameri FA, 2017	PH	PH	DH	DL	PL	PL
Al-Sayaghi KM, 2020	PL	PL	DL	DL	DL	DL
Tabaeian et al., 2017	DL	DL	DL	PL	PL	PL
Al-Khazali et al., 2021	DL	PL	DL	DL	PL	DL
Khalifa et al., 2020	PL	DL	PL	DL	DL	PL
Al-Jaradi et al., 2020	PL	DL	DL	DL	DL	DL
Nahla SA., 2013	DL	DL	PL	PL	DL	DL
Hussein et al., 2020	PL	DL	PL	DL	DL	PL
Yeganeh et al., 2016	DL	DL	PL	DL	DL	DL
Hawsawi et al., 2018	PL	DH	DL	DH	DL	DL
Hassan and Wahsheh, 2017	PL	DL	PL	PL	DL	DL
Aloush and Al-Rawajfa, 2020	DL	PL	PL	DL	DL	DL
Bagheri et al., 2013	PL	DL	DL	PL	PL	DL
Zeb et al., 2018	DL	DH	DL	DL	PL	DL

DL, definitely Low; PL, probably Low, PH, probably high; DH, definitely high.

**Table 4 healthcare-10-01852-t004:** Barriers to adherence to VAP guidelines.

	Aloush, et al., 2017	Atashi et al., 2018	Rashnou et al., 2017	Yaseen and Salameh, 2015	Al-Sayaghi KM, 2020	Alkhazali et al., 2021	Yeganeh et al., 2016	Hassan and Wahsheh, 2017	Aloush and Al-Rawajfa, 2020
Lack of education.	√	√		√		√	√		√
2. Lack of a professional model.	√	√							
3. Poor integration of research findings in practice	√								
4. Unfavourable environmental conditions		√	√						
5. Shortage of nursing staff.		√		√	√	√			√
6. Forgetfulness evidence-based procedures such as wear gloves; sterile technique.					√	√			
7. Lack of policies and protocols.				√	√			√	√
8. Passive human resource and organizational management		√	√						
9. Lack of time		√		√		√		√	
10. Lack of resources		√				√			√

## Data Availability

Not applicable.

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
