# Peer review of "Knowledge, Practice, Compliance, and Barriers toward Ventilator-Associated Pneumonia among Critical Care Nurses in Eastern Mediterranean Region: A Systematic Review"

_healthcare, 2022, doi:10.3390/healthcare10101852_

Round 1
Reviewer 1 Report (Previous Reviewer 2)
Dear authors many thanks for your changes. I have only one consideration: please put more attention to better organising the graphic style of the paper
Author Response
All comments were revised
Reviewer 2 Report (New Reviewer)
The Authors professionally approached an interesting systematic review reacting to information needs concerning knowledge, practice, compliance, and barriers into infection control.
I think this research fills the gap in VAC HAIs in ICU in this WHO region, however there are other systematic review of HCW attitudes towards infection control in the field.
The language and general flow is communicative but in multiple places of the manuscript it was difficult to understand what authors had on their mind. Please consider proofreading service as already in abstract (line 31/32) there are grammatical errors. Please revise also text in lines 187,257/258, 261/262. Language of discussion chapter needs the most corrections.
Authors are quite inconsistent in naming things. I understand that authors took study design directly from the papers (Tab.2) , without redefining, but this is against systematic review principles to summarise knowledge gained from the articles (I have checked it on randomly selected papers). I would suggest to use the list of categories similar to:
-
Experimental: Randomised Control Trials (I've seen 1)
-
Quasi-experimental studies: Non-randomised control studies, Before-and-after study
-
Observational/analytical studies: analytical Cohort study, Case-control study (post intervention comparison), descriptive series of case (convinces based sample of nurses), descriptive cross-sectional (full population of representative sample)
Please ask an epidemiologist to help you, because typical medical professionals are thinking of patients and their diseases as objects, but here a nurse is an object, so the meaning of study design is a little different.
Could authors elaborate more, why they have chosen selected databases for their research? The nursing attitude towards infection control is rather part of public health/ social medicine than clinical research, so most of the surveys or interview related studies are not indexed in selected databases. In the manuscript, studies from KSA, Iran and Jordan are highly overrepresented. Few minute search in Google scholar shows plenty of papers about nurses and VAP from other countries such as UEA, Liban etc. in journals such as International Journal of Nursing Research and Practice. Could authors discuss this in the limitations?
Could authors elaborate more on the classes used to show the most-reported barriers in several studies (tab. 4). I think this is quite important to know what the authors understand under the labels (i.e. Forgetfulness). There is plenty of research about such factors as Bouchoucha, Stéphane L., and Kathleen A. Moore. "Factors influencing adherence to standard precautions Scale: a psychometric validation." Nursing & health sciences 21.2 (2019): 178-185.
The conclusions section is in the obviousness style and not individualised to the presented article. Please rephrase it to indicate interpretation of your results
Author Response
All comments were revised

This manuscript is a resubmission of an earlier submission. The following is a list of the peer review reports and author responses from that submission.
Round 1
Reviewer 1 Report
Introduction needs to be edited and clearly state the research question and identify if is barriers to preventing VAP or reasons for high prevalence of VAP. There is a lot of international evidence on VAP bundles and I am surprised this did not come through in your literature review.
The results table needs to be edited and can be presented clearly. How were the studies graded on quality?
Author Response
Reviewer comment 1 |
Response to Reviewers |
Introduction needs to be edited and clearly state the research question and identify if is barriers to preventing VAP or reasons for high prevalence of VAP. There is a lot of international evidence on VAP bundles and I am surprised this did not come through in your literature review.
The results table needs to be edited and can be presented clearly. How were the studies graded on quality? |
Thanks for your comments. The introduction was improved according to your comments. Kindly, read page 2-3, marked with yellow color.
The table was edited. Kindly, check table 2 for studies graded on quality. |
Reviewer 2 Report
I read with much interest; however, several major revisions are needed. First, the topic is much studied in the literature, so I think it is necessary for the authors to convey the true novelty of the paper. The review claims to be systematic but presents the data in a very descriptive and unclear way. More systematization is needed. It is unclear how the results are aggregated. How do the concepts use for aggregation arise? The steps of the method are sloppily drafted. The discussions are similarly nonspecific and unfocused. The systematic review serves to create the synthesis of new knowledge; as presented, this review does not. Extensive revision of the English language is needed, the abstract needs to be revised (unstructured), and tables that are needed to be fixed.
Author Response
Reviewer comment 2 |
Response to Reviewers |
I read with much interest; however, several major revisions are needed. First, the topic is much studied in the literature, so I think it is necessary for the authors to convey the true novelty of the paper. |
Thank you for your comment , Kindly, you can find the reply for your comments inside the paper marked with yellow color. |
The review claims to be systematic but presents the data in a very descriptive and unclear way. More systematization is needed. |
Thank you for your comment, the paper was more systematization Check pages 2-3 line 70-104 Check page 5, line 186-205. |
It is unclear how the results are aggregated. How do the concepts use for aggregation arise? |
Thank you for your comment, Check table 2 for studies graded on quality. |
The steps of the method are sloppily drafted |
Thank you for your comment, Check page 3 line 107-135 |
The discussions are similarly nonspecific and unfocused. The systematic review serves to create the synthesis of new knowledge; as presented, this review does not. |
Thank you for your comment, Check pages 10-11 |
Extensive revision of the English language is needed, |
Thank you for your comment, Extensive revision of the English language was modified in whole paper |
the abstract needs to be revised (unstructured), |
Thank you for your comment, the abstract was revised check page 1 |
Tables that are needed to be fixed. |
Thank you for your comment, All tables was fixed. |
Round 2
Reviewer 2 Report
Dear authors thanks for the opportunity to review this paper and the change you made.
The overall changes made by the authors were significant, however, there are present some important issues that have to be changed.
Introduction: from lines 102 to 104 the definition of Eastern Mediterranean could be positioned in the introduction (not before the aim) or in the methods to define inclusion and exclusion criteria.
The PRISMA flowchart is completely illegible in the attached format. The summary table.
The attached Table 1 continues to be unclear. In addition to the quality assessment, I do not understand whether the specifications made for the level of knowledge, practice, and compliance is ranked according to the results that emerged or not; the levels indicate how they were obtained. Would it not suffice to indicate the main results section for each element explored?
I would expect the keywords used in the search strategy in the attached table for each database used.
No recording was made on PROSPERO, which generally should be done.
I believe that for these reasons, the manuscript needs to be explored further.
